# Mixed-methods evaluation comparing the impact of two different mindfulness approaches on stress, anxiety and depression in school teachers

Charlotte Todd,[1] Roxanne Cooksey,[1] Helen Davies,[2] Clare McRobbie,[3] Sinead Brophy[1]

[1]College of Medicine, Swansea University, Swansea, UK
[2]College of Human and Health Sciences, Swansea University, Swansea, UK
[3]c/o Swansea University, Swansea University, Swansea, UK

**Correspondence to**
Charlotte Todd;
C.E.Todd@swansea.ac.uk

## ABSTRACT

**Objectives** This study compared the impact of two different 8-week mindfulness based courses (.b Foundations and Mindfulness-Based Stress Reduction (MBSR)), delivered to school teachers, on quantitative (stress, anxiety and depression) and qualitative (experience, acceptability and implementation) outcomes.

**Design** A mixed-methods design was employed. Matched-paired t-tests were used to examine change from baseline, with imputation conducted to account for those lost to follow-up. Qualitative methods involved 1:1 semistructured interviews (n=10). Thematic analysis was used to explore differences in experience between courses.

**Setting** Courses took place in UK primary schools or nearby leisure centres, 1:1 interviews took place via telephone.

**Participants** 44/69 teachers from schools in the UK were recruited from their attendance at mindfulness courses (.b and MBSR).

**Interventions** Participants attended either an MBSR (experiential style learning, 2 hours per week) or .b Foundations (more classroom focused learning, 1.5 hours per week) 8-week mindfulness course.

**Outcome measures** Stress (Perceived Stress Scale), anxiety and depression (Hospital Anxiety and Depression Scale) were evaluated in both groups at baseline (n=44), end of intervention (n=32) and 3-month follow-up (n=19).

**Results** Both courses were associated with significant reductions in stress (.b 6.38; 95% CI 1.74 to 11.02; MBSR 9.69; 95% CI 4.9 to 14.5) and anxiety (.b 3.36; 95% CI 1.69 to 5.0; MBSR 4.06; 95% CI 2.6 to 5.5). MBSR was associated with improved depression outcomes (4.3; 95% CI 2.5 to 6.11). No differences were found in terms of experience and acceptability. Four main themes were identified including preconceptions, factors influencing delivery, perceived impact and training desires/practical application.

**Conclusion** .b Foundations appears as beneficial as MBSR in anxiety and stress reduction but MBSR may be more appropriate for depression. Consideration over implementation factors may largely improve the acceptability of mindfulness courses for teachers. Further research with larger samples is needed.

## BACKGROUND

Mindfulness interventions are currently being used globally in a variety of settings,

### Strengths and limitations of this study

► This study is unique as there are currently no published studies comparing the two mindfulness courses in terms of acceptability, experience and effects on stress, anxiety and depression, despite current roll-out.

► Strengths lie in the mixed-methods approach used to explore differences between .b and Mindfulness-Based Stress Reduction.

► Limitations lie in the numbers lost to follow-up, with future research needed to explore this further.

with a growing interest and uptake in educational settings for both pupils and staff.[1] The specific interest in using schools as a setting for mindfulness intervention reflects a response to the current low levels of well-being and poor mental health experienced by both school children[2] and teachers alike[3] and the call for interventions to address these. Indeed, mindfulness delivered to all in a class setting is one method currently being investigated to help prevent mental health problems in young people without stigmatisation, aiming to improve resilience and coping strategies.[1 4 5] Although many studies are underpowered, early evidence suggests lower stress, depression, improved resilience and improved well-being and emotion regulation for children and young people who undertake mindfulness.[6–9] However, to teach mindfulness in the school setting, teachers are recommended to first learn mindfulness themselves. Research looking at mindfulness for teachers has shown it can reduce stress, burn-out and depression for teachers.[10 11] Given the important role teachers play in the promotion of health in schools, improving their well-being may have positive knock on effects in improving child well-being.[12] However, the small number of studies

included in a recent review of mindfulness based interventions for teachers reflects a need for greater research in this area.[13] Furthermore, with a push to improve well-being in schools, teachers' motivations to learn mindfulness may be primarily centred on teaching it to children, rather than embracing mindfulness as part of their everyday life. Hence, it may be an initiative imposed on teachers, as opposed to something they wish to do for themselves. This means that there may be reduced fidelity and sustainability to the effective mindfulness intervention methods. The courses with the strongest evidence base for mindfulness are mindfulness-based stress reduction (MBSR)[14] and mindfulness-based cognitive therapy,[15–17] in which there is evidence of positive effects on overall health, including mental health benefits. However, the traditional MBSR course is intensive with two or more hours per week (over 8 weeks), and includes sessions such as a guided silent meditation day. This time commitment has been reported to be a barrier to participation in many studies[18 19] and may be a barrier to teachers' participation. In addition, the enquiry component of the MBSR course, which asks participants to reflect on their direct experience of the meditation practice, may not be appropriate for teachers who work closely with each other day to day. Consequently, this has led to the development of modified mindfulness based programmes. In the USA, interventions such as Stress Management and Relaxation Training in Education[20] and Cultivating Awareness and Resilience in Education[21] have been developed for specific use among teachers, administrators and parents of children with special needs. While the majority of the research regarding the teaching of mindfulness to teachers lies within the USA and Canada,[13] there have been studies in the UK.[12] Here, .b Foundations has been specifically adapted for teachers to address their particular needs. This course is shorter (1.5 hours per week for 8 weeks) and there is less enquiry, but more explanation of the scientific evidence and psychology behind the methods and practices. There is a taught or classroom presentation style rather than the MBSR group interactive experiential style with a facilitator. This departure from the traditional evaluated course means it is not clear if the .b Foundations is equivalent to the traditional MBSR and if the same beneficial effect will be seen. As a result, schools face uncertainty over which course teachers should attend. This study set out to compare MBSR with .b Foundations (a teacher adapted mindfulness course) by examining the effects on depression, anxiety and stress levels in school teachers, alongside exploring teachers' experiences of attending the different courses and implementation of teachings into practice.

## METHODS

### Aims and design

A mixed-methods natural experiment evaluation to examine and explore the outcomes and experiences of primary and secondary school teachers who attended either an MBSR course or a. b Foundations course. The specific aims were:

1. To compare the impact of both .b and MBSR on stress, anxiety and depression outcomes in school teachers.
2. To explore any differences regarding the experiences of teachers attending .b and MBSR courses and examine implementation of the courses into teaching and practice.

### Intervention setting

The MBSR course is an 8-week course that last 2 hours each week and is delivered in a group setting. Its nature is experiential and encourages discussion, reflection on direct experience during the course, and homework including 40 minutes of meditation a day. The .b Foundations course runs for 1.5 hours per week for 8 weeks, includes slide show presentations and short practices lasting approximately 20 minutes. All homework is greatly reduced compared with the MBSR course. An outline of key intervention components for each course can be found in table 1. The courses were delivered by school/university teachers who had trained and qualified in MBSR/.b Foundations teaching, were on the teacher training pathway (eg, had supervision, attended mindfulness network conferences and retreats) and had their own personal mindfulness practice. All participating school teachers were offered a free course or a course at a greatly reduced nominal fee . Which course participants could attend was dependent on what was available in their local area. Although participants were not randomly allocated to the interventions, the course that was available in their local area was determined by the funders and so was not influenced by participant characteristics or their individual choice. Courses start dates were between April and September 2016, with different end points.

### Participants

All participants were active teaching staff in primary or secondary schools. They were staff whose school was interested in delivering mindfulness within the school and therefore were encouraged to attend as part of their work and teaching role, or had actively sought to attend a course following hearing about it through their teaching union or other methods, as they personally wished to introduce mindfulness to their teaching. The mindfulness courses were already running and were not arranged as part of this study. Teachers from different schools were mixed within mindfulness sessions and multiple individuals from the same school attended the same course together. Therefore, groups comprised staff who work daily with each other and teachers from other schools who had not previously worked together. At the start of each course, all teachers were handed information sheets and consent forms detailing the study aims, Those that wished to participate, handed in completed forms at the end of the session and provided contact details (where consented) for the researcher to arrange a suitable time for conducting the 1:1 telephone interview. Participants

**Table 1** Intervention components of .b and Mindfulness-Based Stress Reduction (MBSR)

|  | MBSR | .b Foundations |
|---|---|---|
| Length of sessions | 120 min | 90 min |
| Style of teaching | Facilitated group with reflective components and experiential learning and practice. | Taught approach, including slides and presentation. Presentation of scientific bases, evidence and rational for the mindfulness methods. |
| Homework | 30–40 min of formal practice per day and ideas for informal practice. | 20 min of formal practice per day and ideas for informal practice. |
| Practices covered | Mindful movement including yoga and Chi Gong, body scan, breathing practice, sitting practice with duration up to 40 min. | Mindful movement focused on walking, body scan, breathing practice, sitting practice with duration of 20 min. |
| Basis of course | Full catastrophic living by Kabat Zinn. | Finding peace in a frantic world by Mark Williams. |
| Input of the teacher | The course is facilitated and has general concepts in each session but the teacher has a freedom to modify and adapt each session and each course in response to participants. | Structured course. ` |
| Intention of the course | Introduction to a mindful approach to life and tools and practices to cultivate a mindful approach. | Introduction to a mindful approach to life and tools and practices to cultivate a mindful approach. |

were reminded that they did not have to complete part in the research aspect to attend the course and could withdraw from the research component at any time.

## Patient and public involvement
Stakeholders were involved in informing the design of this study as the research question central to this paper arose from engagement with schools and school leadership. Staff at these schools were interested in becoming mindful schools but were uncertain regarding which course teachers should attend in order to enhance well-being in the school. They did not know whether attending the gold standard (MBSR) or shorter teacher adapted version (.b) was better for their staff or pupils. The results of this study will be disseminated to study participants, schools and mindfulness teachers.

## Qualitative methods
From those who wished to participate in the study and consented to be interviewed, 12 participants were purposively selected for a 1:1 semistructured interview based on the course they attended and ensuring representativeness from a male in each group. Two of the participants were not available during the interview period, so a total of 10 interviews were conducted (Five .b participants (four female and one male) and five MBSR participants (four female and one male). Interviews were conducted 3 months after the courses. The topic guide followed a number of questions and prompts in order to evaluate experience of the course, examine implementation of the teachings into practice and explore any differences in experience between those attending .b Foundations and MBSR (box 1). All interviews were conducted by phone by a researcher (CT) experienced in qualitative interviews at a time that was convenient to participants. This researcher had not attended any previous mindfulness

courses or was not involved in mindfulness practice in any way. Participants had not met the researcher as contact was via telephone only. Participants were aware of the research aims through the information sheet given out by the course facilitator during the first week of the mindfulness course. All but one of the interviews was audio recorded and transcribed verbatim and lasted between 13 and 33 min (average 21 min). The remaining participant consented to take part in an interview, but did not want to be recorded. In this case, the participant was happy

**Box 1    Interview schedule**

**Question**
1. Can you tell me about your previous knowledge and experience of mindfulness prior to the course? Prompts: What made you decide to do the course? Were you familiar with mindfulness?
2. Can you tell me about your experience of the mindfulness course? Prompts: What did you find beneficial/unhelpful?
3. Can you tell me if you noticed any changes in your behaviour during and after the mindfulness course? Prompts: Are there other reasons that you know of that could have led to these changes?
4. During the course, can you tell me if and how you used any of the mindfulness practices to help within your teaching? Prompts: did you use any mindfulness practices with the children themselves within class and have you been aware of any effect from doing so?
5. Do you think that you will continue practicing mindfulness/any aspects of mindfulness in the future? Prompts: Could you tell me your reasons for this?
6. What do you think of the mindfulness course as something for other teachers in the future? Prompts: Barriers/facilitators to other teachers engaging if rolled out.
7. Is there anything else that you would like us to know about the experience of the mindfulness curriculum for you personally or the wider school environment?

for extensive notes to be taken and coding undertaken of these field notes.

Two researchers (CT and RC) first read the group of transcripts from .b participants, followed by the group of transcripts from MBSR participants independently. A number of phases of thematic analysis were then conducted following elements of Braun *et al.*[22] Both researchers used open coding to assign a word or phrase to each part of the conversation throughout every transcript. Initial themes emerging were then identified based on these codes and the two researchers met to discuss similarities and differences in coding and theme development. Discourse was also undertaken over any differences between the two groups of participants in terms of codes and emergent themes. This discussion was followed by a final refinement of themes. For example, codes such as 'hippy', 'fluffy' and 'cynical' were all grouped together under 'preconceptions/attitudes towards mindfulness' and whether or not these codes and themes occurred in either .b and MBSR transcripts, or both was discussed. This method of expert validation enhanced the trustworthiness of the research findings.[23]

### Quantitative methods

Participants attending the courses were asked to complete the Hospital Anxiety and Depression Scale (HADS)[24] and the Perceived Stress Scale (PSS)[25] at baseline, end of course and at 3 months follow-up. The HADS is a 14-item scale with seven items related to anxiety and seven related to depression. The PSS is a 10-item scale that measures perception of stress. It is a measure of the degree to which situations in one's life are appraised as stressful. The change from baseline to follow-up for these scores was analysed using matched-pair t-tests, so that change for the individual was reported using exploratory analysis to give confidence intervals around the mean difference for respondents. Gender and age were adjusted for in this analysis. Multivariate imputation by chained equations in STATA (version 14) using baseline and end of course data was used to impute follow-up data for those lost to follow-up. Data were assumed to be missing at random (eg, probability of being missing does not depend on the missing value). When imputation is used, the sample size is larger and so SEs will be reduced.

## RESULTS

Forty-four out of 69 teachers from both primary and secondary schools in two local authorities were recruited from their attendance at four different courses (two .b and two MBSR). The .b courses had 83% (15/18 (4 males)) and 90% (18/20 (1 male)) attendance of at least 6 out of the 8 sessions, respectively, with reasons given for drop-out including requiring a foot operation, timing of year, school events and busy schedules (Headteacher). The MBSR courses had 71.4% (15/21 (4 males)) and 100% (10/10 (1 male)) attendance of at least 6 out of 8 courses, respectively, with reasons given for drop-out or non-attendance including after school responsibilities (running after school club, events for parents and school trips), lack of childcare and hospital admission for injury.

Of these 69 attendees, 44 (64 %) consented to participate in the quantitative research aspect and completed baseline questionnaires. There were 17 (1 male: 16 female, average age 42.5 (range 28–61 years) participants who attended the .b Foundations courses and 27 (5 male: 22 female, average age 40 (range 24–58 years) who attended the MBSR courses. Baseline differences between participants on the course showed those attending the MBSR had higher levels of depression, stress and anxiety at baseline compared with those attending .b Foundations (see tables 2–4). Baseline scores for stress and depression were not statistically different for those lost to follow-up.

**Table 2** Changes in depression score between baseline, end of intervention and follow-up (Hospital Anxiety and Depression Scale)

| Course | Depression score (baseline) (mean±SD) | Depression score (end of intervention) | Depression score (follow-up) | Imputed depression score (follow-up) | Mean difference from baseline (complete data only) | Mean difference from baseline (imputed data) |
|---|---|---|---|---|---|---|
| .b | 4.76±3.68 (n=17) | 2.58±1.88 (n=12) | 3.9±3.7 (n=10)* | 3.54±3.2 (n=15) | 0.1 (95% CI −0.47 to 0.62) | 0.86 (95% CI −0.76 to 1.76) |
| MBSR | 6.78±3.08 (n=27) | 3.47±2.70 (n=19) | 3.44±3.61 (n=9)† | **2.88**±3.4 **(n=20)** | **4.4 (95% CI 0.75 to 8.13)** | **4.3 (95% CI 2.5 to 6.11) n=20** |
| Difference | **2.02 (95% CI 1.56 to 2.48)** | | | | | |

0–7=normal, 8–10=borderline abnormal (borderline case), 11–21=abnormal (case).
*Baseline score in the 10 with follow-up was 4.0.
†Baseline score in the 9 with follow-up was 7.9.
MBSR, Mindfulness-Based Stress Reduction.
Bold values are statistically significant at p<0.05.

**Table 3** Changes in anxiety score between baseline, end of intervention and follow-up (Hospital Anxiety and Depression Scale)

| Course | Anxiety score (baseline) (mean±SD) | Anxiety score (end of intervention) | Anxiety score (follow- up) | Imputed anxiety score (follow-up) | Mean difference from baseline (complete data only) | Mean difference from baseline (imputed data) |
|---|---|---|---|---|---|---|
| .b | 10.47±3.64 (n=17) | 5.83±3.21 (n=12) | 7.1±4.04 (n=10)* | 7.1±4.1 (n=15) | **2.8 (95% CI 0.06 to 5.5)** | **3.36 (95% CI 1.69 to 5.0)** |
| MBSR | 11.89±3.50 (n=27) | 7.65±3.23 (n=20) | 7.66±3.84† (n=9) | 7.83±3.7 (n=20) | **4.78 (95% CI 1.52 to 8.03)** | **4.06 (95% CI 2.6 to 5.5)** |
| Difference | 1.42 (95% CI 0.93 to 1.9) | | | | | |

0–7=normal, 8–10=borderline abnormal (borderline case), 11–21=abnormal (case).
*Baseline score in the 10 with follow-up was 9.9.
†Baseline score in the 9 with follow-up was 12.4.
MBSR, Mindfulness-Based Stress Reduction.
Bold values are statistically significant at p<0.05.

## Qualitative findings

Analyses were first done separately to explore any difference between participants' experiences attending .b Foundations and MBSR courses, however, themes were similar across both groups and no major differences were apparent. Consequently, results are presented as four overarching themes covering experiences from both .b and MBSR participants. A table has been constructed with further illustrative quotes from participants attending both courses (table 5):

### Preconceptions/attitudes towards mindfulness

Preconceptions and beliefs regarding mindfulness were the major theme throughout the analysis. Participants discussed this in relation to their own preconceptions, as well as their opinions about others' attitudes towards mindfulness including teachers, children and parents. In relation to teachers' preconceptions, many commented on their prior beliefs as 'sceptical' and 'cynical', but noted how these attitudes and beliefs changed throughout the course:-

I was a bit sceptical to start with, because I'm not a very sort of sit still type of person, but yeah, you know,

all I can say is it works, so if it works, you can't knock it. (Participant I, MBSR)

The meditation aspect of mindfulness was remarked on in relation to this theme, with one participant noting that they did not even like to use the word meditation or let others know that that they were engaging in such an activity:

The whole idea of meditating, I actually always thought was quite a sort of a hippyish kind of thing, if you know what I mean…Flowery and fluffy… I never thought I would be the kind of person that would embrace meditating, although I actually kind of don't even like using that word, I tend to say 'I'm just going to go and get some sort of headspace. So I think, yeah, it did sort of change my perceptions of it a little bit, yeah. (Participant C,. b)

The notion of other teachers' preconceptions' was brought up by a few participants, particularly when discussing widening teacher engagement in mindfulness:

It's also the, you know, the image, when people talk about mindfulness they think, 'oh so like stuff'…like

**Table 4** Changes in perceived stress between baseline, end of intervention and follow-up (Perceived Stress Scale)

| Course | Stress score (baseline) (mean±SD) | Stress score (end of intervention) | Stress score (follow-up) | Imputed follow-up score | Mean difference from follow-up | Mean difference from baseline (imputed data) |
|---|---|---|---|---|---|---|
| .b | 20.35±8.72 (n=17) | 13.08±7.54 (n=12) | 15.7±6.36 (n=10) | 13.0 (n=15) | 2.8 (95% CI –2.6 to 8.15) (n=10) | 6.38 (95% CI 1.74 to 11.02) |
| MBSR | 22.52±5.54 (n=27) | 14.25±5.22 (n=20) | 16.22±8.6 (n=9) | 12.3 (n=20) | 6.1 (95% CI –2.0 to 14.2) (n=9) | **9.69 (95% CI 4.9 to 14.5)** |
| Difference | **2.17 (95% CI 1.17 to 3.16)** | | | | | |

*Baseline score in the 10 with follow-up 18.5.
†Baseline score in the 9 with follow-up 22.3.
MBSR, Mindfulness-Based Stress Reduction.
Bold values are statistically significant at p<0.05.

**Table 5** Illustrative quotes from Mindfulness-Based Stress Reduction (MBSR) and .b participants

| | .b Foundations | MBSR |
|---|---|---|
| Prior awareness | Some: All participants had some prior knowledge or awareness | Some: Majority had some prior knowledge of mindfulness, one had no prior knowledge |
| Reason for attending | Personal benefit (2/5) | Personal benefit (3/5) |
| | Pupil benefit (1/5) | Pupil benefit (1/5) |
| | Both personal and pupil benefit (2/5) | Headteacher led (1/5) |
| Experience | Group as positive (5/5) | Group as positive (4/5) |
| | You get talking to people and you know we would share practice or share experiences and that was a valuable thing too so I think it was nice to do things in a group, and also from a personal point of view and, I have, you know with the best intentions of having doing mindful practice every day but I don't always do as much as I should, and because I knew I was going to group, it made you. (Pt D) | if you were to do that by yourself, then you would probably walk away, but the fact is there are other people in there, and those people reinforce what had gone on in our activity, and they said that they'd reuse it, and it had had a positive effect, so that does encourage you then to perhaps try it for yourself, not to get too, 'Oh, I'm not sure if this is going to work, or if this is going to be right for me,' you know. (Pt G) |
| | Setting is important: needs to be convenient with minimal distraction | Setting is important: needs minimal distraction |
| | it was in a school I was working in so I didn't have to go and get in the car and go anywhere so I never missed any of the sessions. On the other hand, it was in a school hall that's normally used for an after school club so on the nights that the course was on the after school club were moved into a classroom next door and you could sometimes hear the noise of the children which was a little distracting in the beginning (Pt C) | it's very important 'cos it was quite quiet, there were a couple of occasions when there was some drama going on upstairs, some practice, drama practice, they moved, you know, but I think it was nights they were voting and there was a lot of noise and I think it is important that you have a quiet space (Pt H) |
| | Facilitator as school teacher: excellent | Facilitator as passionate in subject: excellent |
| | the facilitator, was, excellent and, you know, she's obviously a real expert, she said it's been part of her life for donkeys' years and so she was, you know, she really understood what the barriers were and was able to sort of guide us through those weeks where you're saying I'm, not getting anywhere with this and, you know, I'm not doing what I should be doing (Pt A) | [Name of facilitator] was fantastic, you could see [facilitator] was passionate about the subject they were teaching (Pt G) |
| Barriers to attendance | Felt preconceived ideas will affect others attendance | Felt Preconceived ideas will affect others attendance |
| | So they were a bit more black and white about it, it's sort of all a bit fluffy for them maybe, I don't know if this is the right terminology or, you know, but I think that it's a little bit uncomfortable for them. (Pt A) | um, there's a sort of attitude with some people that it's got to be, it's a very sort of pure kind of Zen-like activity and you've got to do it in a particular way, you've got to be sitting in a particular way (Pt F) |
| | Time perceived as too long (1.5 hours) | Time perceived as too long (2 hours) |
| | They were quite long, I would say an hour and a half is quite a long time to be in school extra so I think that would put some people with families off (Pt E) | with school, it is hard to know, you know, an hour a week would be much easier for everybody, so perhaps the next step could be an hour a week and more people being able to commit to it, and if that's, if you're talking about teachers signing up to it, you know, I know out of experience (Pt H?) |
| Perceived impact | Calmer | Calmer |
| | I'm so much calmer, so much calmer and just I think I'm more patient with the children in my class and everything and I'm just a bit more, less controlling really (Pt B) | I'm more aware of when I'm getting anxious.when my breathing was getting a bit out of control and I'm getting uptight, I spot that much sooner and I feel I've got strategies I can use to sort of calm things down a bit (Pt F) |
| | Less reactive | Less reactive |
| | recognise the way I'm reacting to situations as well, you know, the trying to kind of think about, think you know, Paws.b, you know the kind of like, take those moments to you know reflect on what's happening and just make me a little bit calmer rather than reacting to situations, having that moment of reflection (Pt E). | so I've put that into practice, again just taking that sort of deep breath, not reacting to things as quickly, just taking a moment to think about it, and yeah, that's sort of what I've been putting into practice (Pt I. |
| | Moreaware | More aware |
| | I feel I'm much more able to deal with situations where I see my colleagues getting really wound up and I stay really calm, and then they say to me, 'How do you stay so calm?' And I really think the mindfulness has helped me with that (Pt D). | having done the course it sort of reminded you to sort of take that step sort of back and live in the moment because I think that because we have busy lives you're so sort of thinking of, 'right, what have I got to do next, what have I got to do next?", and you miss what you're actually doing (Pt I) |

Continued

**Table 5** Continued

| | .b Foundations | MBSR |
|---|---|---|
| Commonly used techniques | Breathing techniques | Breathing techniques |
| | I use the short, the sort of. b, the stop and think, you know the take a breath and calm down, that sort of thing. I use those for myself all the time when I'm dealing with children who are being aggressive or you know if because they're angry about something and it's helped me really to just stay calm (Pt D) | when I feel stressed I go, right, slow down, stop, think about your breathing, but I haven't made time in my daily routine really for the actual meditation which I know would help a lot (Pt H) |
| | Body scan | Body scan |
| | one thing that they do where you're sort of going through the body, you're just very quiet and still and you focus on your breathing and you just kind of then through your body sort of literally just checking each part of the body, sort of rolling down, do you feel any discomfort, anything in your head and it's done through you sort of looking through your body and I find that really relaxes me, so that's one I've used quite a lot. The other thing is…(Pt C) | the body scan exercise within the mindfulness is quite similar, and that, that I've started using a bit more of on myself as a means of sort of like at the end, especially at the end of the day when I go to bed now it's body scan and it really does sort of like relax you to have a good night's sleep (Pt I) |

I think some men in school would think oh what a load of rubbish, you know what I mean, you know, they wouldn't want to be seen to be engaging with it (Participant H, MBSR)

Indeed, one participant pointed out that using a 'pure' mindfulness approach with teachers could potentially be a barrier to engagement, placing emphasis on the importance of discussion and dialogue for people in this profession. Another expressed the need for scientific rationale and evidence behind engaging in mindfulness as a facilitator to teachers' participation:

I think I sort of had this attitude that it was all a little bit airy-fairy, hippy-dippy sort of nonsense, but to hear that there's you know sort of proper scientific evidence of how it does, you know, change your brain… I think, you know, for a lot of people you want to know that there's some concrete research rather than just sort of anecdotal evidence (Participant A,. b)

One teacher suggested that changing the name of the course might be more appealing for teachers who may already have preconceived ideas regarding what mindfulness entails:

Mindfulness sounds all being all mindful and being all… I don't know…, like if you called it something… like dealing with stress, yeah, stress at work, as opposed to mindfulness, where people think, oh you know, away with the fairy's type thing. (Participant H, MBSR)

This belief regarding others attitudes was not limited to teachers, but stemmed across to parents and children, with teachers proposing that parental, as well as child beliefs may influence any future delivery of mindfulness to children. A few teachers relayed their experiences of practising such an approach with children, commenting on children's' initial reactions to the idea of mindfulness:

They all thought I'd cracked, you know, I'd gone…. some thought that I was just burning into a mad witch with magic potions, you know. (Participant H, MBSR)

### Factors influencing teachers' experiences of attending mindfulness course

All teachers (except one MBSR participant) had some knowledge of mindfulness and commented on different reasons for attending the courses. Most had been initially invited to attend by school leadership staff or the National Union of Teachers. The extent to which leadership staff encouraged school staff to attend was not known, so it is difficult to gauge the pressure felt by teachers to attend the courses. Of the MBSR participants, three mentioned their interest was first for themselves, one mentioned their reason was mainly for teaching children and one commented that it was compulsory from their Headteacher. With the .b Foundations participants, while many mentioned the reason for attending was initially for themselves, four also mentioned a second reason was for the benefit of their pupils.

Most teachers interviewed discussed teaching as a factor, if not the main contributor to their stress levels, with only one participant (Participant J) explicitly stating it was not their main source of stress. A lack of understanding from higher authority and government was also expressed as a source of stress, with teachers noting a need to understand the reasons behind the stress experienced by teachers:

It is a really stressful job. I'm not sure that people outside the profession know how stressful it is and I don't know that government realise how stressful it is and I don't think anybody's ever stopped to have a look to find out why it's so stressful. (Participant F, MBSR)

Under this theme, a number of subcategories came up relating to teachers experience, including the group, the setting, the course facilitator and the timing.

### The group situation
The group situation was a major subcategory central to teachers' experience of the mindfulness course, with all participants commenting on factors relating to being part of a group. One participant went as far as saying that being part of a group with shared interests and backgrounds was

possibly the most positive part of the mindfulness course and would be a large facilitator to engaging teachers:

> what was the most positive, and in a way, you know, not really linked to the mindfulness was just that we were a group of teachers coming with sort of common concerns and that was really, really good. I think we all got a lot out of that…it was a positive coming together of people from a, with a common sort of background of stress and anxiety linked to the work…because there was some dialogue and a chance to talk, I think that would make it more beneficial for teachers as a profession I think (Participant F, MBSR)

Indeed, many remarked on the group as a facilitator to sustained engagement. While some discussed the group as facilitative, others suggested encountering some barriers to such an approach, mainly related to the group consisting of people whom they worked closely with.

Only one participant had an overall negative experience of the group setting as they felt it could turn into a bit of 'a talk shop' for people, which they felt was a hindrance, remarking that they did not enjoy discussion with people they did not know well. Overall, the group situation was reflected on as a highly positive part of most teachers' experience.

### The setting

Another subcategory frequently discussed in relation to teachers' experience was the setting of delivery. The setting was either at a school (in some cases participants' own school) or at a local leisure centre. Those participants who attended at their own school talked about the pros and cons of this situation, discussing both the convenience of being at their own school but also the distractions associated with this:

> it was convenient in one way because for me it was in a school I was working in so I didn't have to get in the car and go anywhere so I never missed any of the sessions…On the other hand it was in a school hall that's normally used for an after school club…, and you could sometimes hear the noise of the children which was a little distracting. (Participant C,. b)

Overall, the need for minimal distraction seemed to be the most important factor about the setting.

### The course facilitator

The person delivering the mindfulness was felt to be particularly important by participants, particularly in being able to guide them through the practice with awareness of the barriers they may face. The positives of one of the course facilitators being a school teacher was also reflected on, as well as the belief that the course facilitator needs to be open and adaptable for the group they are working in:

> As an ex-teacher herself, it felt like she really knew where we were coming from… yeah, for me just sort of relate to her more and you felt that she could

relate to you and knew where you were coming from… I think it was particularly useful that she was an ex-teacher. (Participant C,. b)

> put teachers in a room and, you know, they can't help but chat with one another, so I think the benefits of us being teachers together could be reduced if the mindfulness tutor was more in that sort of strict frame of mind (Participant F, MBSR)

### The time and length of course delivery

Participants from both .b and MBSR seemed to be happy with the timing of the course and felt the course was a good duration. While some commented to feel worried about the time required to begin with, they also noted that they felt the benefits of the course outweighed the time invested:

> to begin with it did feel like "oh gosh, it's another thing to fit in…I think the time is why I haven't done it as a daily practice myself, it's purely the kind of fitting it in but I think that the value of it outweighs the time that you need to put into it. (Participant C,. b)

With regard to any differences in viewpoints regarding the timing of the courses between .b and MBSR participants, one MBSR participant did feel that you would not get the same benefits from a shorter course:

> two hours wasn't it…that was just about right really, you wouldn't want any longer but any shorter I don't think you'd have the benefit really. (Participant H, MBSR)

Despite this, this same participant also remarked on the difficulty in getting teachers to engage in a 2-hour session when discussing widening course access:

> with school, an hour a week would be much easier for everybody…I've tried with teachers to do a (name of class) after school, well they certainly didn't want to do a two hour one, so we've managed to get an hour and a half class after school, out of the evaluation they've felt an hour and a half was too long so an hour was what suited them. (Participant H, MBSR)

Indeed, when asked about sustaining practice and engaging other teachers in mindfulness, the majority of participants mentioned time as a main barrier and the difficulty in getting people to understand the benefits of commitment.

### Perceived impact of mindfulness courses

Many participants seemed surprised regarding the impact they had noticed from attending the 8-week course. This was from both a work and personal perspective. One particular participant commented on improved resilience and changes to well-being:

> I feel it's had a huge impact on my emotional wellbeing, I feel very much more resilient and calmer and just generally happier I think, I can't tell you how

different it's made my outlook on things and, you know the stresses at work, I feel totally different about things… a much, much, much bigger impact than I thought it would have, there's all kinds of stresses in my personal life as well that I'm feeling more resilient about.(Participant A,. b)

Many participants commented on the impact the course has had on allowing them to feel more calm, less reactive and more aware of situations (table 5) and more at ease with taking 'space' out of a busy day. Breathing techniques were discussed among practical strategies used regularly:

> the sort of calming down, the breathing and sort of getting control of myself was one of the things that I really learnt. So, when things are getting a bit stressful or even when things are just really busy at work, I'm able to find five min and just sit quietly in a quiet room somewhere and just get myself back on an even keel really. So I think that would be my most positive thing, that it did give time to find those spaces in a really busy week. (Participant F, MBSR)

Participants discussed the impact they felt the course had on their teaching and school life, with some noting a difference to their interactions with children including greater patience, being less controlling and experiencing reduced tension in stressful situations:

> I think it's helped in my teaching practice and dealing with, as I say these challenging children. You know I can cut them more slack as well, I mean I was always a patient person but this made me even more patient and just more realistic really and much calmer generally you know, and able to deal, I feel I'm much more able to deal with situations where I see my colleagues getting really wound up and I stay really calm, and then they say to me, 'How do you stay so calm?' And I really think the mindfulness has helped me with that. (Participant D,. b)

However, throughout analysis, some possible confounders and factors influencing participants' perceived impact of mindfulness were discussed such as seasonality, previous knowledge, interests, and mindfulness being a part of an overall lifestyle change.

> I think the mindfulness was part of a bigger set of strategies I was trying to put in place…so the mindfulness has just been part of that but I think it's been a really good part of that (Participant F, MBSR)

However, one participant ruled out possible confounders, attributing mindfulness as the main reason for feeling calmer:

> it's definitely that (mindfulness) because we've had our school inspection and I'm planning a wedding at the moment so I shouldn't be calm (Participant B,. b)

Despite the many positive impacts noted, a few participants felt that mindfulness was just a mechanism to manage stress and not actually get to the root cause.

> I don't think it's made me, it hasn't made the job any less stressful but I think it has made it more manageable. (Participant F, MBSR).

Participants remarked on a need to use a bottom up approach to reduce the sources of stress in teaching:

> we're putting a band aid on a bullet hole, if you want people to genuinely feel happier in our society, then you genuinely have to change it, from the bottom up, I think mindfulness is a strategy for helping people deal with the contemporary modern day problems, which is overwork, a sense of futility, perhaps… mindfulness is just another mechanism for burying ones head in the sand, and ignoring the fact that there is a lot of things, on our planet, that really need people to take to the streets to solve, but at the same time, it's given me more progressive and holistic strategies. (Participant G, MBSR)

For these reasons, participants felt that mindfulness would work for some people or certain types of people, but not all.

### Ongoing training desires and practical application with children

After participating in the course, four out of five MBSR participants and three out of five .b Foundations participants commented they had used mindfulness based approaches with children in some capacity. All but one expressed that they would like to use mindfulness in schools. Seven participants specifically expressed a desire to go onto the paws .b course to deliver mindfulness to children and one expressed interest in becoming a mindfulness school. The other two participants mentioned the potential value of teaching children but did not clearly state that they would go on to attend any further courses. Reasons expressed for wanting to progress to teach children including an increase in emotional problems among children and a need to help address this as well as stay calm on a personal level:

> The thought of doing it with children was quite appealing, you see more and more children coming into schools now with emotional and social difficulties, children with sort of rising stress levels.(Participant C,. b)

A few participants expressed a lack of confidence to teach children without attending the paws .b course. Despite this, some did mention that attending the course had made them calmer in their interactions with children, possibly acting as a mechanism of indirect impact. Reflective of this, a few participants emphasised the need for teachers to be calm in order to impact on the children:

> I think if we're to be role models for the children and develop healthy, confident children then we need to

be sort of quite well balanced in ourselves so I think it would be good if in teacher training they found some sort of time to put a little bit of this in. (Participant C,. b)

Some teachers had used mindfulness-related techniques in the school setting with children, commonly discussing use of breathing techniques but would like to gain further skills in this area of application with children.

### Quantitative findings

Participants completed measures of depression, stress and anxiety at baseline, end of course and after 3 months follow-up. Of 44 participants who completed baselines, 19 (43%) also completed the follow-up questionnaire. Using only the data from participants who completed the follow-up data showed that in these participants, MBSR was associated with a reduction in depression and anxiety compared with baseline (see table 1) and .b Foundations was associated with a reduction in anxiety. Accounting for those lost to follow-up by using imputed data modelled on baseline and end of study data to estimate follow-up scores, MBSR was associated with reduction in anxiety, stress and depression and .b Foundations associated with reductions in anxiety and stress but not depression.

### DISCUSSION

This study has found that both mindfulness courses (MBSR and .b Foundations) are associated with reductions in stress and anxiety for teachers. There was no evidence that the teacher adapted .b Foundations course is associated with reduction in depression. However, the MBSR course was associated with improved depression outcomes.

In terms of qualitative experiences of the mindfulness courses, there were no major differences between participants attending .b or MBSR courses. Participants who undertook the MBSR course did not report to find it less feasible or enjoyable compared with those who undertook the .b Foundations course. A number of themes emerged throughout analysis, which were common to participants attending both courses, a major one being that of preconceptions and prior beliefs regarding mindfulness as an approach. A number of participants from both .b Foundations and MBSR commented on how one's ideas and beliefs regarding mindfulness and the word meditation itself can act as a potentially large barrier to initial take up of a mindfulness course, particularly among teachers. Other research has also shown misconceptions and perceptions as a main barrier to meditation related practice[26] and blog authors have reported similar perceptions stating they initially thought that mindfulness meditation was for hippies, but following direct experience found benefits from the practice.[27] While the potential benefits of mindfulness are published in the scientific world, greater emphasis and communication of evidence regarding the potential benefits and consideration over

changing the name of the course, could go some way towards acceptability for teachers. This need for evidence was also brought up by University staff participating in a workplace mindfulness intervention and the significance participants attached to knowing the intervention was intellectually sound was believed to be a core part of enabling a key stage of change (resonance). Part of this resonance also related to the feeling of belonging to a group and acceptance,[28] another highly important facilitator to the intervention reported in our study.

Both MBSR and .b Foundations courses have a group structure, which was found to be highly facilitative to the majority of participants, particularly as the group consisted of people within the same profession, experiencing similar stressors. The majority of participants recognised the benefits of group work including; supporting and sustaining the practice, and discussing shared challenges and interests, however, one teacher did find working closely with ones' peers (who were in a higher role/authority to themselves) sometimes inhibited discussion and another felt there was a bit too much talking. Unequal participation and intrinsic conflict in same professional groups are known barriers to group formations,[29] and perhaps should be viewed as an inevitable part of the process, as the overall feeling was that the group method of delivery was a highly beneficial, and perhaps even an integral part of the process.

Other factors found to be influential and important to the acceptance of the course were the venue and person delivering the course. According to literature,[30] serious consideration for the appropriateness of the setting (venue) for group work should include; appropriate size (for comfort)—not so large as to intimidate group participants or inhibit adequate group cohesion, accessible (to all) but removed from immediate 'business', to minimise the risk of undue interruption. Therefore, on reflection, future courses should consider a venue which is in close proximity of the school (but perhaps not in the school itself), which encompasses a relaxed and comfortable atmosphere, with minimal distraction. The positive impact of the course facilitator encountered by participants in this study and perhaps a contributor to the fact that no major differences were found between experience of MBSR and .b Foundations courses, could be influenced by the fact that both mindfulness facilitators had teaching elements to their employment experience. Perhaps this enhanced the experience for participants and such benefits may not be seen from facilitators without a teaching background. This may be a crucial aspect to be considered for delivery of future mindfulness courses. With participants' believing that they may have not experienced these benefits had a strict inward looking tutor delivered the course, there is an argument that a course facilitator with good understanding and experience of the particular professional group to which mindfulness is being delivered to goes a large way towards improving acceptability. Should the course facilitator not have this understanding, this acceptability may be largely reduced.

Another factor commented on by many participants in relation to their experience of the course was timing and duration of course delivery. One main difference between the .b Foundations is the length of individual sessions. One MBSR participant felt that you would not get the benefit of the course if sessions were under 2 hours. However, as nearly all participants expressed time as a potential barrier to engagement, and no differences were found in terms of the level of stress reduction obtained through both courses, the .b Foundations course may potentially be more acceptable and feasible, particularly if one aim is to engage more teachers to go forward to deliver mindfulness to children. However, further research is needed to establish if some of the effectiveness or quality may be lost if implemented on a larger scale.

Participants in both courses reported subjective benefits in terms of patience and improvements in stress and tension. They also commonly discussed feeling calmer, more aware and less reactive. While this study did not specifically set out to explore mechanisms by which each programme may work in lowering stress, many of these perceived impacts resonate with the stages of change proposed from mindfulness interventions[28] Furthermore, managing emotional reactivity is a key part of emotion regulation, which may act as a key mechanism of change in lowering stress and improving well-being.[13 31 32] Indeed, a pilot study[11] examining the stress, burn-out and teaching efficacy found significant reductions in psychological symptoms and burn-out following a modified MBSR course adapted specifically for teachers. However, some participants felt mindfulness was just a method of coping with stress rather than tackling the causes of stress. This idea is reiterated in social blogs discussing mindfulness in the workplace, suggesting it may be more of a tool to help cope with stress, when ideally the work environment needs to change.[33] Perhaps greater emphasis needs to be placed on the root origins of the stress experienced by teachers today and higher level policy and practice changes may be needed to produce truly sustainable reductions in stress. This may go some way to addressing the issue, however, it could be argued that due to individual differences in responses to stress,[34] even at low levels of stress, there will still be a need and a potential benefit from intervention for some individuals.

Teachers in this study perceived themselves as 'doer's' 'perfectionists' with ongoing to-do-lists, which is polar opposite to the underpinning philosophy of mindfulness which embraces 'being', 'acceptance' and 'non-striving' as an integral part of its practice. Some teachers associated meditation as 'self-indulgent' and were concerned about the responses of their peers, parents and children of their school. However, teachers noted that, many children, on becoming more familiar with mindfulness, became more accepting of the practice and would engage with breath works and finger breathing, and even requested more mindfulness. A need to focus on the early years as a preventative approach was also expressed by many participants. With high numbers from both courses articulating

a desire to go on further training to teach children, both courses seem to be successful at gaining the platform level of experience and knowledge needed to interest participants in attending future course and rolling out delivery to children. Some participants felt that attending the course changed their approach to dealing with pupils and a few already felt confident to use mindful-based practices with children. This known, it would be interesting to establish whether purely training teachers in mindfulness on a personal basis has a positive impact on child outcomes, or whether the delivery of mindfulness directly to pupils brings the larger benefits. Indeed, while mindfulness-based interventions are increasing in popularity and the evidence base is beginning to build with children,[35–37] research in this area is still in its infancy and particularly scarce among very young children, the age group perceived by participants in this study as those who would benefit most from such intervention.

### Study limitations

This study was not a randomised controlled trial, so participants were not randomised to the interventions. The baseline data showed that those who undertook the MBSR had slightly more depression, anxiety and stress and this may be because the name of the course being stress reduction appealed to participants who felt more stressed or made participants more willing to report their stress more than those who participated in the .b Foundations course. However, the fact that these participants may have been more stressed to begin with and possibly self-selected on to courses also highlights these individual differences in vulnerability to stress and anxiety exist[34] and self selection on to courses should not be seen as purely negative. Indeed, these programmes may just have an additional benefit or work better for teachers who feel they are in need or feel they could benefit from such courses.

The teachers who took part in the mindfulness courses expressed views that it was possibly seen as embarrassing to be engaging with 'mindfulness' and there were notably fewer men taking part in the courses. Thus, the views and experiences of those who did not take part in the course may be very different from those who did participate and these findings may not be transferable to people who did not choose to do a mindfulness course. The facilitators in both courses were people with teaching backgrounds and the style of the MBSR course will have been based on their teaching style. It is possible that MBSR courses delivered by those who do not have a style which is tailored to school teachers (eg, that with a counsellor, mental health or Buddhist style) would be less acceptable than we found in this study. Some of the courses also ended very close to the end of summer term when participants are potentially likely to be less stressed anyway. However, sustained improvements in stress reduction were shown at follow-up when participants returned to school, limiting the confounding influence of seasonality. Another possible limitation is the lack of control for potential confounders

such as other lifestyle changes, which could result in improvements in stress and anxiety such as increased uptake of physical activity. It is difficult to determine if other lifestyle changes such as these, happened before the mindfulness started or whether the mindfulness encouraged this uptake of positive lifestyle changes

## CONCLUSION

Teachers participating in both .b and MBSR courses showed significant improvements in stress and anxiety at both end of intervention and 3-month follow-up and those attending MBSR courses showed significant improvements in depression scores at these time periods. This suggests that .b Foundations is as beneficial as MBSR in terms of anxiety and stress reduction but MBSR may be more appropriate for depression. Further research with larger samples is needed to explore these findings further.

While no major differences were found in this study comparing those who undertook MBSR with .b Foundations, in terms of acceptability or experience, a few notable points can be taken forward for future delivery of mindfulness-based courses for teachers. The type of course, being MBSR or .b may not be the key point influencing acceptability for teachers and in fact, greater importance may be placed on other factors. The course facilitator being knowledgeable and having experience regarding the professional group to which they are delivering mindfulness to and the course facilitator being adaptable to the groups' particular needs seem to be key factors. Having a dialogue and a chance to share ideas and practice with people of a similar background was highly important for teachers and not recognising this may be detrimental to the success of future courses. Furthermore, with nearly all participants discussing preconceptions and time as a major barrier for teachers, shorter sessions perhaps over a longer time period and consideration over the name under which courses are advertised may largely improve acceptability if attempts are made to roll out mindfulness to school teachers on a larger scale. However, careful process evaluation should be undertaken to ensure fidelity is not lost.

**Acknowledgements** All authors would like to thank the teachers attending mindfulness courses for their cooperation during the study and specifically those participating in research outcomes.

**Contributors** The study was designed by CT, HD, CM and SB. Data collection was performed by CT. Quantitative data analysis was performed by SB (blinded), whilst qualitative analysis was performed by CT and RC. The first draft of the paper was written by CT and all authors provided critical input and revisions for all further drafts. All authors have read and approved the final manuscript.

**Funding** Funding to deliver the mindfulness courses was received from the National Union of Teachers (NUT). For evaluation, the work was undertaken with the support of The Centre for the Development and Evaluation of Complex Interventions for Public Health Improvement (DECIPHer), a UKCRC Public Health Research Centre of Excellence. Joint funding (MR/KO232331/1) from the British Heart Foundation, Cancer Research UK, Economic and Social Research Council, Medical Research Council, the Welsh Government and the Wellcome Trust, under the auspices of the UK Clinical Research Collaboration, is gratefully acknowledged. The work was also undertaken with support of the Austin Bailey Foundation and the National Centre for Population Health and Wellbeing Research (NCPHWR).

**Competing interests** Three of the authors in this paper are mindfulness practitioners and have a strong prior belief in the benefits of mindfulness. This could be perceived as a potential conflict of interest. However, this is a study comparing two different methods of teaching mindfulness. The motivation for this research was to examine if .b Foundations and MBSR were equivalent or if one course was clearly better in a school environment. Indeed, one of the authors teaches both MBSR and .b Foundations, showing no bias to any particular method of teaching. These authors were not involved in the analysis of qualitative work or interpretation but contributed to the methods (how mindfulness was delivered) and design. One researcher who is also a mindfulness practitioner undertook the quantitative analysis, but was blinded to the course identity.

**Patient consent for publication** Not required.

**Ethics approval** This study was approved by the University College of Human and Health Science Ethics Committee (reference number: 10416).

**Provenance and peer review** Not commissioned; externally peer reviewed.

**Data sharing statement** Further quotes from qualitative aspects used to generate codes and themes can be shared from the lead author on request.

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
