## [Reviewer comments · BMJ Open]

ARTICLE DETAILS

TITLE (PROVISIONAL)	A mixed methods evaluation comparing the impact of two different mindfulness approaches on stress, anxiety and depression in school teachers
AUTHORS	Todd, Charlotte; Cooksey, Roxanne; Davies, Helen; McRobbie, Clare; Brophy, Sinead

VERSION 1 - REVIEW

REVIEWER	Dr Siobhan Hugh-Jones University of Leeds, UK
REVIEW RETURNED	05-Nov-2018

GENERAL COMMENTS	This is a clear paper, offering pragmatic information about the most appropriate type of mindfulness training for teachers. I suggest the following minor revisions prior to publication: - greater reference in the introduction to the global work on mindfulness in schools, and where the UK is in relation to that. The paper may give the false impression that .b Foundations is the only teacher focused programme.- clarification on whether participants were primary or secondary school teachers- outline of key intervention components. These could be tabulated. Without this, it is difficult to know that the interventions are comparable in terms of content and hypothesised mechanisms of change.- reporting the interview questions posed (these clearly shaped the data)- were people asked about extent of home practice?- were differences at baseline controlled for?- why was no measure of mindfulness administered? This should be considered in the study evaluation. Without this, can you claim that the mindfulness component of the intervention led to outcomes?- greater reference in the discussion to mechanisms of change, ie. did teachers experience the course in the way intended? As I know them well (!), I am pointing you to two papers that report many of the experiences your sample did. It would be good to see more nuanced consideration of the group effect as well as the need for evidence, cited by your participants. https://link.springer.com/article/10.1007/s12671-017-0691-4 https://link.springer.com/article/10.1007/s12671-017-0790-2
---

	- understandably, your analysis had to be limited; the study evaluation could comment on the fact that you did not explore in depth the ways in which teachers felt that mindfulness had helped to lower stress, anxiety and depression (and not just that it had). Further differences between the two courses may have been identified here. - I would suggest that it is not necessarily a weakness that participants were likely to be motivated in ways that others teachers are not. This motivation is appropriate when starting a course, and simply suggests that these programmes can work for people who feel a need / are motivated. - the argument about mindfulness being a sticking plaster is well noted. However, one could argue that even if systemic stress were reduced, there would still be individual differences in perceived stress, coping skills and resources, as well as vulnerability to anxiety and depression - hence there might always be a need for interventions like these.
--	--

REVIEWER	Aniko Biro Institute of Economics, Centre for Economic and Regional Studies of the Hungarian Academy of Sciences Hungary
REVIEW RETURNED	04-Dec-2018

GENERAL COMMENTS	It is not clear why teachers are the most suitable population to compare MBSR and .b Foundations. The Authors mention the issues of time commitment and working closely with each other, but I do not see why these issues would be specific to teachers. If the Author's compare the impact of the two methods among teachers because their primary interest lies in the implementation of the courses into teaching practice then they should state so. What is the reason for not considering online MBSR programmes as alternative intervention? Online training could particularly be convenient for teachers, considering the above mentioned issue of time constraints. How exactly did the recruitment take place? Please provide details. To impute missing values of follow-up data, the Authors use multivariate imputation by chained equations. Considering that the analysis is based on a small baseline sample (N=44) and the rate of attrition is large (57%), I am worried that the imputation method leads to attenuation bias in the standard errors (as is suggested by the results presented in Tables 1-3). Please discuss. Also, exact details of the imputation method are needed. When presenting the qualitative findings, the authors provide interesting insights into the views of the participants. However, as there were only 10 participants in the qualitative study, all of whom self-selected into the mindfulness course, it is difficult to figure out what scientific evidence could be taken away from these details. Maybe a table summarising the most frequently raised positive and negative aspects of the training could serve as a guidance for the readers. In particular, a comparison of the responses from .b Foundations and MBSR participants would be helpful.
--

	I am worried about the following citation: "it was possibly slightly difficult doing it with so many people that I work with, particularly because (name of school lead) was doing the course as well, so sometimes you sort of felt that you couldn't say things that you might have said had you not known anyone there...on the other hand, it probably made it more comfortable going with people that I work with and that I know" Because of the few participants in the study, and the supposedly low participation from school leaders, I'm afraid that this citation could potentially violate the anonymity of the respondent. As the Authors acknowledge as limitations, the participants were not randomised into .b Foundations or MBSR treatment, and there was no control group. Because of these limitations, it is not clear what the readership could learn from the current analysis. The beneficial effects of mindfulness on depression, anxiety and perceived stress have already been documented in the literature based on more refined research designs. An added value of the study could be the comparison of .b Foundations and MBSR trainings, but that would require randomisation into the treatment, and a larger sample size. Also, the implementation of the trainings into teaching practices turn out not to be part of the analysis. Because of these considerations, I think the manuscript does not merit publication, at least not in its current form.
--	---

VERSION 1 – AUTHOR RESPONSE

Reviewers' Comments to Author:

Reviewer: 1

Reviewer Name: Dr Siobhan Hugh-Jones

Institution and Country: University of Leeds, UK Please state any competing interests or state 'None declared': None declared

This is a clear paper, offering pragmatic information about the most appropriate type of mindfulness training for teachers. I suggest the following minor revisions prior to publication:

Thank you for your helpful comments and suggestions to improve the clarity of the manuscript.

Please find amendments below.

- greater reference in the introduction to the global work on mindfulness in schools, and where the UK is in relation to that. The paper may give the false impression that .b Foundations is the only teacher focused programme.

The introduction has now been added to give greater context with further evidence drawn on in the main manuscript.

- clarification on whether participants were primary or secondary school teachers

Participants were both primary and secondary school teachers. This is written in the participants section but has also been added into the methods section of the manuscript under aims and design to give further clarity.

- outline of key intervention components. These could be tabulated. Without this, it is difficult to know that the interventions are comparable in terms of content and hypothesised mechanisms of change.

See Table 1 :MBSR compared to .b Foundations.

- reporting the interview questions posed (these clearly shaped the data)

This has been added into the main manuscript as Table 2.

- were people asked about extent of home practice?

Reflections on home practice is an important part of the MBSR course and this is part of every session in MBSR. People were asked about their practice in the interview and most commonly reported practices are incorporated into an added table showing illustrative quotes (Table 6).

- were differences at baseline controlled for?

Yes, differences such as gender and age were adjusted for in the analysis. This has been added into the main manuscript.

- why was no measure of mindfulness administered? This should be considered in the study evaluation. Without this, can you claim that the mindfulness component of the intervention led to outcomes?

This paper is not intending to measure if mindfulness 'works', it was to examine the experience of participants on the two different courses and if one course was associated with better outcomes compared to the other course. We were evaluating the courses rather than mindfulness.

- greater reference in the discussion to mechanisms of change, ie. did teachers experience the course in the way intended? As I know them well (!), I am pointing you to two papers that report many of the experiences your sample did. It would be good to see more nuanced consideration of the group effect as well as the need for evidence, cited by your participants.

<https://link.springer.com/article/10.1007/s12671-017-0691-4>

<https://link.springer.com/article/10.1007/s12671-017-0790-2>

Greater reference to this has been added in to the discussion and new table (Table 6).

- understandably, your analysis had to be limited; the study evaluation could comment on the fact that you did not explore in depth the ways in which teachers felt that mindfulness had helped to lower stress, anxiety and depression (and not just that it had). Further differences between the two courses may have been identified here.

Authors acknowledge this, whilst it was not the specific aim of this study to explore mechanisms behind each course, greater attention has been given to this in the paper and also the new table which has been added in whereby feeling calmer, being less reactive and more aware are key changes reported and we have reported illustrative quotes for each. Reference to this has also been included in tracked changes in the main manuscript.

- I would suggest that it is not necessarily a weakness that participants were likely to be motivated in ways that others teachers are not. This motivation is appropriate when starting a course, and simply suggests that these programmes can work for people who feel a need / are motivated.

Authors acknowledge this and address the point in the main manuscript.

- the argument about mindfulness being a sticking plaster is well noted. However, one could argue that even if systemic stress were reduced, there would still be individual differences in perceived stress, coping skills and resources, as well as vulnerability to anxiety and depression - hence there might always be a need for interventions like these.

Authors acknowledge this and address in track changes of the main manuscript.

Reviewer: 2

Reviewer Name: Aniko Biro

Institution and Country: Institute of Economics, Centre for Economic and Regional Studies of the Hungarian Academy of Sciences, Hungary Please state any competing interests or state 'None declared': None declared

It is not clear why teachers are the most suitable population to compare MBSR and .b Foundations. The Authors mention the issues of time commitment and working closely with each other, but I do not see why these issues would be specific to teachers. If the Author's compare the impact of the two methods among teachers because their primary interest lies in the implementation of the courses into teaching practice then they should state so.

Authors acknowledge that the paper could benefit from greater clarity and context. The introduction has been added to in the main manuscript to give greater rationale as to why teachers were chosen as the most suitable population.

.b Foundations can only be taught to teachers. The people who learn to teach .b Foundations agree to not teach this to the general public or to anyone outside the teaching profession. It is a course that has been designed specifically for teachers in a schools or higher education environment.

What is the reason for not considering online MBSR programmes as alternative intervention? Online training could particularly be convenient for teachers, considering the above mentioned issue of time constraints.

The programmes evaluated were what was currently on offer and available for teachers. Whilst online training may be an appealing approach for teachers, evaluated online mindfulness programmes specific for teachers were not available at the time of this paper.

How exactly did the recruitment take place? Please provide details.

Details have been provided in the participants section of the main manuscript.

To impute missing values of follow-up data, the Authors use multivariate imputation by chained equations. Considering that the analysis is based on a small baseline sample (N=44) and the rate of attrition is large (57%), I am worried that the imputation method leads to attenuation bias in the standard errors (as is suggested by the results presented in Tables 1-3). Please discuss. Also, exact details of the imputation method are needed.

We have undertaken imputation in order to examine the influence the loss to follow up will have on the findings. We have presented our imputation results alongside our observed results. This enables the reader to see in impact of loss to follow on the study findings. The reader can see both observed and imputed data so can come to their own conclusions.

Authors do not feel attenuation bias is relevant here. Attenuation bias is due to measurement error - it is when the correlation coefficient is weakened because of measurement error in the exposure variables. There is no measurement error in the exposure variable here as it is attended .b course or MBSR course. We did not impute for course attendance as this was fully observed. We have fully observed records for the exposure variables.

When presenting the qualitative findings, the authors provide interesting insights into the views of the participants. However, as there were only 10 participants in the qualitative study, all of whom self-selected into the mindfulness course, it is difficult to figure out what scientific evidence could be taken away from these details. Maybe a table summarising the most frequently raised positive and negative aspects of the training could serve as a guidance for the readers. In particular, a comparison of the responses from .b Foundations and MBSR participants would be helpful.

A table presenting key findings has been added to the main manuscript (Table 6) to provide greater information which could be useful to readers in this area. Not all participants self-selected on to the course, some were told by their headteachers and some were doing purely for benefit of pupils rather than personal benefit. This study helps to inform future directions in introducing mindfulness in schools as it is the first direct comparison of the main courses used in the UK in schools.

I am worried about the following citation: "it was possibly slightly difficult doing it with so many people that I work with, particularly because (name of school lead) was doing the course as well, so sometimes you sort of felt that you couldn't say things that you might have said had you not known anyone there...on the other hand, it probably made it more comfortable going with people that I work with and that I know" Because of the few participants in the study, and the supposedly low participation from school leaders, I'm afraid that this citation could potentially violate the anonymity of the respondent.

This quote has been removed in the main manuscript to avoid any likelihood of identification of the respondent.

As the Authors acknowledge as limitations, the participants were not randomised into .b Foundations or MBSR treatment, and there was no control group. Because of these limitations, it is not clear what the readership could learn from the current analysis. The beneficial effects of mindfulness on depression, anxiety and perceived stress have already been documented in the literature based on more refined research designs. An added value of the study could be the comparison of .b Foundations and MBSR trainings, but that would require randomisation into the treatment, and a larger sample size. Also, the implementation of the trainings into teaching practices turn out not to be part of the analysis. Because of these considerations, I think the manuscript does not merit publication, at least not in its current form.

The participants were not randomised. We agree this was a real world research and in schools teachers often do the course that is locally available to their schools. This study did not set out to do a trial. There was no control group because we were not evaluating mindfulness, we were comparing two courses. Mindfulness vs controls has been evaluated many times.

The merit of this research is it is the first study to compared MBSR and .b Foundations. These are the two main courses used in the roll out of mindfulness in schools in the UK. The formation of the All Party Parliamentary Group in UK government show the importance this has to the country and this information is informative in the decision regarding roll out. This is the first evaluation comparing the programmes to use in introducing mindfulness in schools.

VERSION 2 – REVIEW

REVIEWER	Siobhan Hugh-Jones University of Leeds, UK
REVIEW RETURNED	27-Jan-2019

GENERAL COMMENTS	I believe the authors have addressed my comments clearly and that the paper is now suitable for publication.
--

REVIEWER	Aniko Biro Institute of Economics, Hungarian Academy of Sciences
REVIEW RETURNED	29-Jan-2019

GENERAL COMMENTS	The Authors addressed most of the issues I raised before. I have a few remaining remarks: I welcome the addition of Table 6, but in its current form it is difficult to overview. Could the Authors make the table more concise? I still miss details on the imputation method. What are the assumptions that ensure the imputation and the standard errors are valid?
--

VERSION 2 – AUTHOR RESPONSE

Thank you for these comments in helping to improve the manuscript. I have added information on these aspects in the tracked changed document and made the table more concise , which improves readability.